# The Effect of Diversification under Different Ownership Structures and Economic Conditions: Evidence from the Great Recession

**Ivonne A. Liebenberg [1,\*] and Zhilu Lin [1,2]** 

[1] Department of Finance, School of Business Administration, University of Mississippi, Oxford, MS 38677, USA; zlin@clarkson.edu

[2] The David D. Reh School of Business, Clarkson University, Potsdam, NY 13699, USA

\* Correspondence: iliebenberg@bus.olemiss.edu

**Abstract:** The effect of corporate diversification on firm performance has been extensively documented in the literature. In the general finance literature, Kuppuswamy and Villalonga (2015) studied the diversification effect during the 2007–2009 financial crisis and found that diversification adds value in the presence of external financing constraints. Motivated by this finding, we investigate whether a similar effect applies to insurance firms and we develop hypotheses for their different ownership structures (stock vs. mutual insurers; and group vs. non-group affiliated insurers). Using a sample of property-liability insurers over a period of 2004 to 2013, we find that the effect of diversification on performance is contingent on ownership structures and economic conditions. The diversification effect for stock insurers and insurers affiliated with a group is not significantly affected by economic conditions. However, the diversification effect for mutual insurers and non-affiliated insurers is reversed during the financial crisis. More specifically, diversified firms with these kinds of ownership structures perform better than focused firms during normal economic conditions, but their performance was significantly worse during the financial crisis. Our results are robust to alternative measures of performance and diversification, and to corrections for endogeneity. Our study contributes to the diversification literature by showing how the effect of diversification varies with ownership structure under different economic conditions and the results shed light on the specific circumstances in which diversification can improve or reduce performance.

**Keywords:** corporate diversification; financial crisis; Great Recession; property-liability insurance

## 1. Introduction

There is an extensive literature on the value of corporate diversification. Corporate diversification is associated with both costs and benefits and its average net effect is largely an empirical question. The value of corporate diversification is contingent on the time period in which is measured, the geographic location, the data used to measure it, and the statistical methods implemented (Villalonga 2004). Stein (2003) concludes that it is more useful to focus on the variation of the value of diversification rather than its mean value.

Some studies focus specifically on investigating the effects of diversification under different economic conditions. Kuppuswamy and Villalonga (2015) and Rudolph and Schwetzler (2013) study the value of diversification during the 2007–2009 global financial crisis, also known as the Great Depression. They find that diversification adds value during the financial crisis and argue that corporate diversification can provide an important insurance function for investors. Kuppuswamy and Villalonga (2015) conclude that internal capital markets are more valuable and more efficient when



external capital becomes more expensive. They argue that the value of diversification increases during the financial crisis due to two effects that are not mutually exclusive but may complement each other[1]. The "more money" effect suggests that when capital is rationed, creditors prefer to lend their scarce funds to safer conglomerates than to riskier focused firms. The "smarter money" effect implies that the relative value of internal capital markets increases when credit constraints are binding.

Kuppuswamy and Villalonga (2015) and Rudolph and Schwetzler (2013) study only public firms and exclude the financial sector. Motivated by their finding, we investigate whether the increased value of diversification during the financial crisis extends to insurance companies. Their samples only contain stock firms, thus whether their hypothesis applies to firms with different ownership structures remains an open question. We believe that the value of diversification and the effects of the financial crisis on the value of diversification may be contingent on the different ownership structures of insurance companies. The costs and benefits of diversification should be different in different ownership structures and they should be influenced differently by the financial crisis.

In our study, we examine the diversification effects in the property–liability (P/L) insurance industry under different ownership structures, mutual vs. stock firms, and groups vs. non-groups firms, and under different economic conditions. Our full sample is comprised of 7470 firm-year observations from the National Association of Insurance Commissioners (NAIC) database for the years 2004–2013. We use ROA and ROE as a proxy for firm performance and a two-stage least square (2SLS) and a Heckman model to control for a potential endogeneity problem.

We find that the effect of diversification on performance is contingent on ownership structures and economic conditions. Diversified stock insurers and group insurers have a similar performance to non-diversified stock and group insurers, regardless of the economic conditions. Thus, the effect of diversification for these kind of ownership structures is not significantly affected by economic conditions. However, the diversification effect for mutual insurers and non-affiliated insurers is reversed during the financial crisis. Diversified mutual and non-group insurers perform better than focused firms during normal economic conditions, but their performance was significantly worse during the financial crisis. Our results are robust to alternative measures of performance and diversification.

The financial crisis provides a natural environment for researchers to test the diversification effects in the presence of external financing constraints and when the expectation of performance falls. Ivashina and Scharfstein (2010) indicate that new loans to large borrowers fell substantially during the financial crisis period. Also, most businesses experienced a reduction in performance during the financial crisis period. Stein (2003) claims that during a financial crisis period, diversified firms may have larger internal capital market and use their capital more efficiently than focused firms.

It is valuable to explore the diversification effects in the insurance industry since it is a highly regulated industry in the financial sector, and the premium growth of the P/L insurance industry declined significantly during the financial crisis period due to price declines and reduced economic activity (Harrington 2009) (See Figures 1–5). One of the largest bailouts that happened during the financial crisis was an $85 billion federal government bailout of American International Group (AIG), which is a well-diversified multinational insurance corporation. Moreover, mutual firms and stock firms have coexisted in the insurance industry for a long time. Different characteristics are inherent in stock firms and mutual firms. Since in a mutual company the policyholders are also the owners of the firm, it is harder to monitor managers in a mutual company than in a stock company (Mayers and Smith 1981). Stock firms have the advantage of access to external capital. The diversification effects of stock firms and mutual firms will be different because of those different characteristics. Similarly, we explore the diversification effects for group-affiliated insurers and those who are not affiliated. Group insurers have access to a larger internal capital markets than non-affiliated insurers. Thus, we expect to see a difference in the diversification effect between these two types of insurers.

---

[1]    These two effects were originally introduced and labeled by Stein (2003).

Our contribution to the diversification literature is threefold. First, we add the study of the diversification effects in the insurance industry in the presence of extreme financial conditions to the literature and benefit from the richness of insurance statutory data. Second, we not only measure the effect of diversification for stock firms (as in the rest of the finance literature), but also add the study of the diversification effects for mutual firms, firms affiliated with a group, and non-affiliated insurers. Third, some researchers find that the diversification discount may be due to the biases related to the COMPUSTAT database (Villalonga 2004). According to Liebenberg and Sommer (2008), we benefit from the insurance statutory data in that it eliminates measurement error and managerial discretion in segment reporting.

Our study has important practical implications for firms and regulators in the insurance industry. (1) Firms should take into account their ownership structure when deciding whether to diversify or not and to what extent. (2) Firms should take into account the economic cycle in which they are and the expected economic activity when deciding whether to diversify or make an adjustment to their current diversification status. This is specifically important for mutual insurers and non-group insurers as our results show that diversification can reduce performance for these firms in the presence of external financial constraints. (3) Firms and regulators should encourage alternative governance mechanisms for mutual and non-group insurers. During economic downturns and situations in which external capital is restricted, these alternative governance mechanisms could help to more closely monitor these type of ownership structures and provide more guidance to help them to undergo the crisis.

The remainder of this paper is organized as follows. Section 2 summarizes the existing literature. Section 3 introduces hypotheses development. Section 4 describes the data and methodology. Section 5 presents the results, and Section 6 summarizes our conclusions.

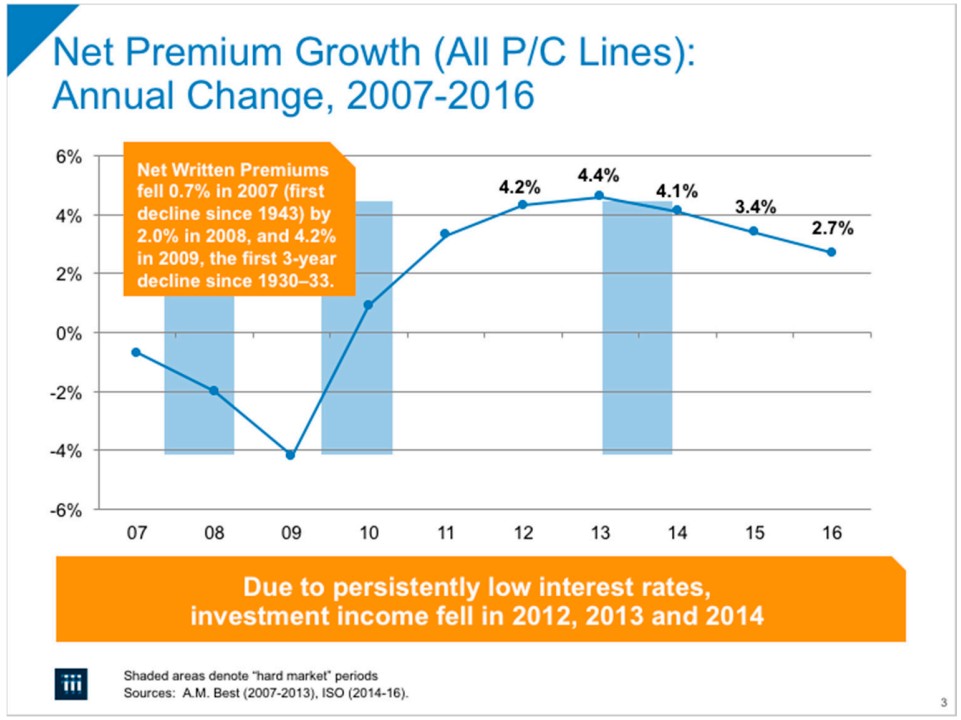

**Figure 1.** Net Premium Growth (All P/C Lines): Annual Change, 2007–2016.

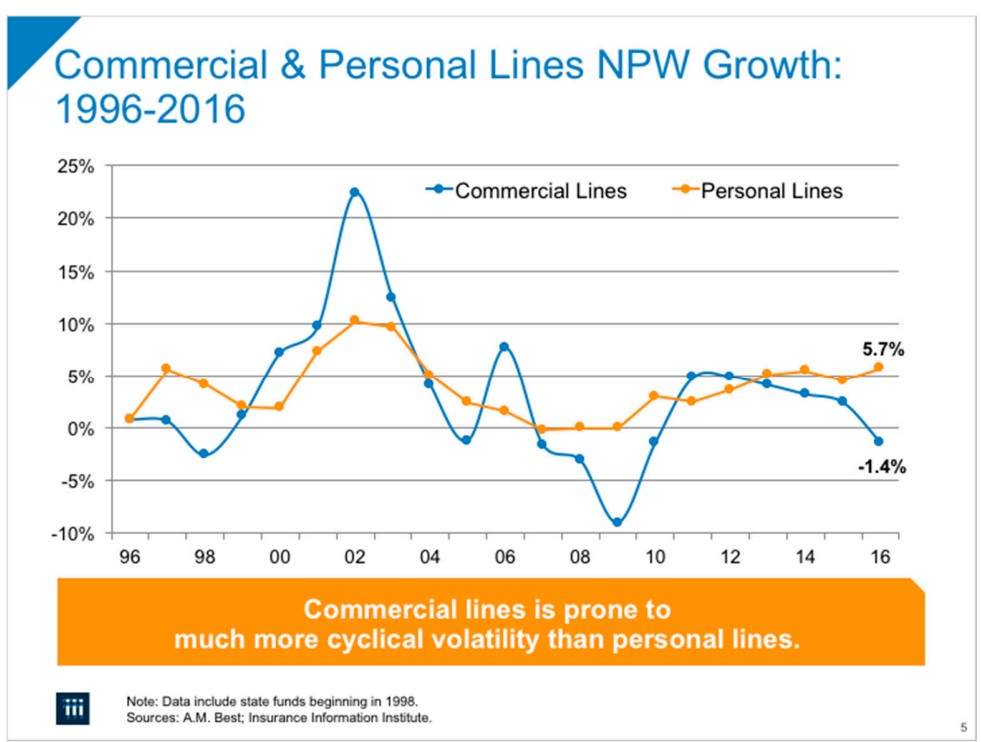

**Figure 2.** Commercial & Personal Lines NPW Growth: 1996–2016.

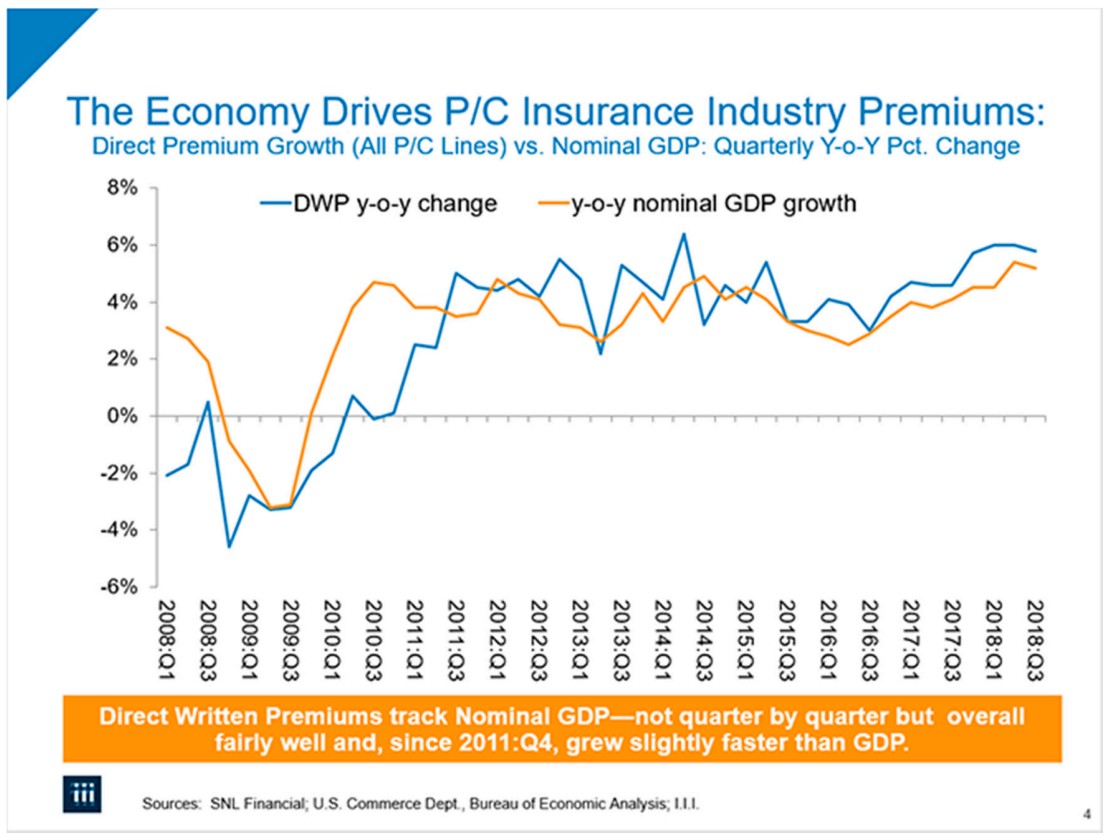

**Figure 3.** The Economy Drives P/C Insurance Industry Premiums.

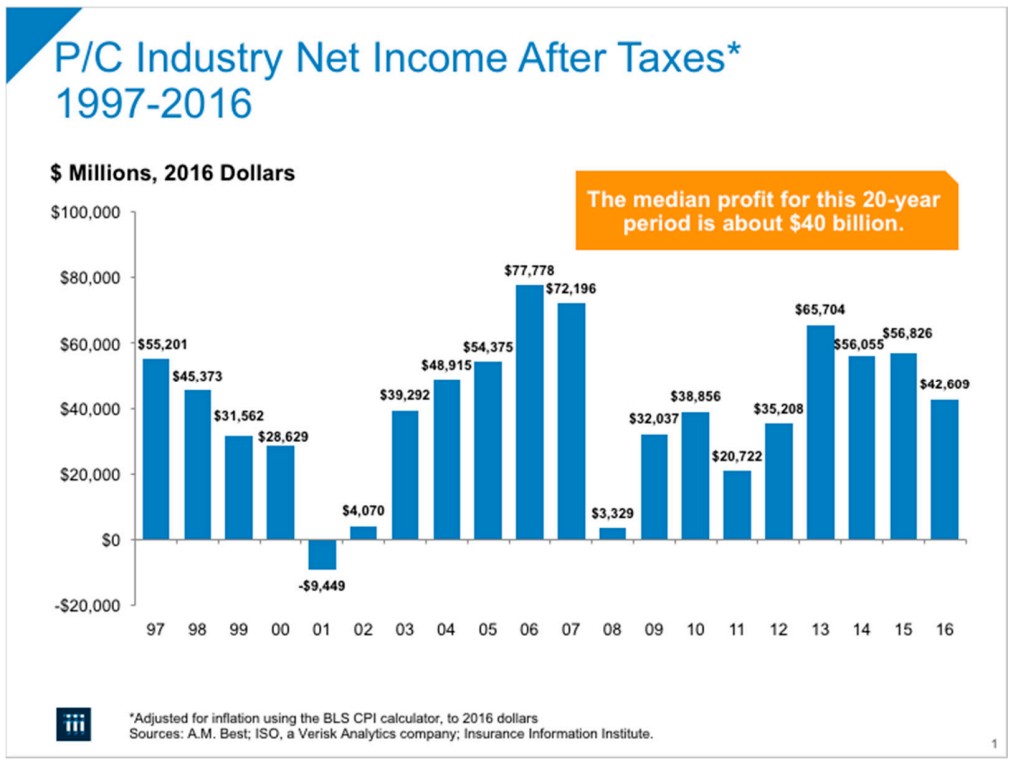

**Figure 4.** P/C Industry Net Income After Taxes, 1997–2016.

## P/C Insurance Industry Investment Income

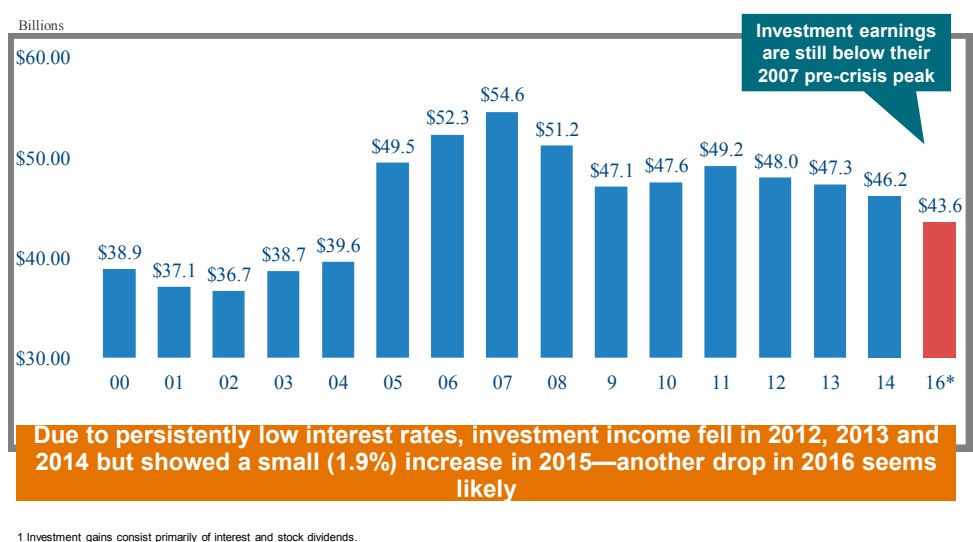

**Figure 5.** P/C Insurance Industry Investment Income.

## 2. Previous Literature

The benefits of diversification include debt coinsurance, efficient internal capital markets (ICM), use of non-tradable resources, economies of scope, and increased market power. The costs of diversification include inefficient investment and agency problems such as managers' personal risk reduction, empire building, and managerial entrenchment. These benefits and costs and the associated literature are summarized in Appendix A and described in more detail below.

*Benefits of Diversification.* The benefits associated with diversification are a larger efficient internal capital market, increased debt capacity, and economies of scope. The larger internal capital market generated by diversification may be more efficient than external capital. Using internal capital can reduce the transaction costs associated with the sale of securities to the public and the cost of overcoming information asymmetry problems encountered when selling securities in the capital market (Hadlock et al. 2001; Myers and Majluf 1984). Stein (1997) documents that corporate headquarters can reallocate funds toward promising projects that might be capital-constrained if pursued within stand-alone firms. Moreover, diversification also enhances a firm's debt capacity by reducing the overall firm's risk through combining businesses whose cash flows are less than perfectly correlated (debt coinsurance effect proposed by Lewellen (1971)). Customers are willing to pay higher insurance premiums from companies that are less risky (Herring and Santomero 1990; Sommer 1996; Cummins and Danzon 1997). Finally, diversified firms are able to share fixed production costs across several businesses within the firm and transfer firm-specific intangible assets such as brand name and customer loyalty (Teece 1980; Markides 1992).

*Costs of Diversification.* The costs of diversification are inefficient internal capital allocation and exacerbation of managerial agency problems. Diversified firms create a large internal capital market and if they use the internal capital to subsidize poor performance segments, diversification may lead to a value loss. Berger and Ofek (1995) find that inefficient cross-subsidization exists in internal capital markets. Furthermore, when firms become larger and more complex, monitoring managers becomes more difficult, so diversified firms face greater agency problems. For example, segment managers may make decisions without considering the financial situation of the whole company, they may overinvest, or even participate in perquisite activities to favor their own interest. Ozbas and Scharfstein (2009) find that the inefficient investment behavior of conglomerate firms is at least in part due to agency problems. Denis et al. (1997) document that firms maintaining value-reducing diversification strategies suffer from agency problems.

In the general diversification literature, there are two conflicting hypotheses that explain the possible effects of diversification on firm value or performance. The conglomerate hypothesis emphasizes the benefits associated with diversification and suggests a positive relationship between firm performance and diversification. The strategic focus hypothesis, on the other hand, emphasizes the costs associated with diversification and suggests a negative relationship between firm performance and diversification. Villalonga (2004) finds a diversification premium using a census database supporting the conglomerate hypothesis. However, the prevailing literature suggests that diversification destroys firm value, supporting the strategic focus hypothesis. For example, Hoyt and Trieschmann (1991) study publicly traded insurers that specialize in either P/L or L/H insurance and those that diversify across different segments and find that specialized insurers have higher risk-adjusted returns compared to diversified insurers for the period 1973–1987. Tombs and Hoyt (1994) use a Herfindahl index of premiums written across 10 business line groups to measure diversification and find that insurers that are more diversified have lower stock returns. Cummins and Nini (2002) study P/L insurers and find that diversified insurers have a lower accounting performance than focused insurers. Similarly, Liebenberg and Sommer (2008) examine P/L insurers over the period 1995–2004 and find that diversified insurers have lower accounting performance and market-based performance than focused insurers.

In summary, the value of corporate diversification is the average net effect of its benefits and costs and the question of whether diversification adds or reduces value is an empirical question. Stein (2003) suggests that it is more useful to focus on the variation of the diversification discount or premium than on its mean value. Studies have shown that the effect of diversification is contingent on the data and methodology used, geographic location, time period, etc. More specifically, the existing literature finds that the effect of diversification is affected by institutional environments, economic stability, and affiliation with business groups (Chakrabarti et al. 2007; Mitton 2002; Rudolph and Schwetzler 2013). Moreover, researchers show that the discount may be due to selection bias, endogeneity, and the biases related to the COMPUSTAT database (Graham et al. 2002; Campa and Kedia 2002).

Some studies have focused specifically on investigating the effects of diversification under different economic conditions. Dimitrov and Tice (2006) document that diversified firms perform differently and relatively better than focused firms during recessions. More recently, Kuppuswamy and Villalonga (2015) and Rudolph and Schwetzler (2013) find that the diversification discount declines during the 2007–2009 financial crisis. Kuppuswamy and Villalonga (2015) conclude that diversification adds value in the presence of external financial constraints because internal capital markets are more valuable and more efficient when external capital becomes more expensive. The "more money" effect suggests that when capital is rationed, creditors prefer to lend their scarce funds to safer conglomerates than to riskier focused firms. This effect implies that the coinsurance feature of conglomerates should reduce their default risk and increase their debt capacity. The "smarter money" effect implies that the relative value of internal capital markets increases when credit constraints are binding.

## 3. Hypotheses Development

In this section, we develop hypotheses explaining how the costs and benefits of diversification (and the resulting net effect) may differ for different ownership structures and under different economic conditions in the insurance industry.

### 3.1. Ownership Structures under Normal Economic Conditions

Stock insurers and mutual insurers have coexisted in the insurance industry for a long time. The two major differences of the two organization types are agency problems (Mayers and Smith 1982, 1988) and access to external capital. First, owner–policyholder conflicts and owner–manager conflicts are two main incentive conflicts in the insurance industry (Mayers and Smith 1981). In a mutual insurance firm, the policyholders are also the owners of the firm. This is in contrast to a stock organization in which owners and customers are separated. Stock firms face owner–policyholder conflicts and owner–manager conflicts. Therefore, the mutual form reduces the policyholder-owner agency problem, but this reduction in owner–customer agency costs may be offset by greater owner–manager agency costs because monitoring managers is difficult in mutual firms. Second, stock insurers can access external capital, but mutual insurers cannot access external capital. Mutual insurers have the capital from their policyholders, and they can issue surplus notes and other debt capital.

We hypothesize that, under normal economic conditions, the effect of diversification on performance is contingent on the insurer's ownership structure. The diversification effects in these two types of organizations could be different because of the differences in agency problems and accessibility to external capital. The benefits of diversification, increased debt capacity and access to internal capital markets, are more important for mutual firms than for stock firms. Mutual insurers can only raise capital from policyholders and issue surplus notes and other debt capital. Stock insurers have more channels to raise capital (such as issuing stocks) than mutual insurers. Therefore, mutual insurers will benefit more from increased debt capacity and access to a larger internal capital market than stock insurers. However, the cost of diversification, agency problem, is greater in mutual firms as they are not as closely monitored as stock firms. Since the magnitude of the diversification benefits and costs should be different in mutual firms and stock firms, we examine the diversification effect in mutual firms and stock firms separately.

We also examine the diversification effect in firms affiliated with a group and those that are not affiliated, separately. We hypothesize that under normal conditions, group-affiliated insurers have access to larger internal capital markets through their groups. The diversification benefit of access to internal capital markets for firms that are affiliated with a group are then redundant. However, this benefit of diversification is more important for non-affiliated insurers as they do not have access to a group's internal capital markets. Therefore, the relative value of an internal capital market is higher for non-group than for group insurers.

### 3.2. Ownership Structures under Extreme Economic Conditions

The economic conditions changed drastically during the Great recession. Credit and external capital became more expensive and more difficult to obtain. Thus, it is highly unlikely that the overall benefits and costs of diversification during the crisis were the same as those before crisis.

The effect of diversification largely depends on whether the internal capital allocation is efficient. Stein (1997) analyzes under what circumstances the benefits of internal capital allocation are most likely to exceed its costs. He finds that it is precisely when credit constraints are binding. Under such economic conditions, individual projects within the firm are forced to compete for the scarce funding, and the incentive of headquarters to choose the most deserving project is also increased. Yan (2006) also finds that the value of conglomerates increases compared to focused firms when external capital is more expensive at the aggregate level. Kuppuswamy and Villalonga (2015) find that diversified firms use their internal capital more efficiently during the financial crisis (smarter money effect). Additionally, using diversified conglomerates data, Matvos and Seru (2014) demonstrate that some firms reallocate resources internally to significantly mediate the effect of financial shock. However, Mitton (2002) focuses on the East Asian financial crisis of 1997–1998 and finds that, while diversification can offer the benefit of improving capital allocation, this benefit could virtually disappear in a time of crisis as investment opportunities diminish. Moreover, if agency problems rise during a financial crisis period, internal capital allocation will become inefficient for diversified firms. During the financial crisis period the expectation of firm performance fell. Agency problems will be more severe in diversified firms when the expectation of firm performance falls since monitoring managers is more difficult in diversified firms than focused firms and the incentive of manager's perquisite activities will increase. Johnson et al. (2000) argue that managers tend to expropriate minority shareholders during a crisis as the expected return on investment falls.

According to the debt coinsurance theory, diversification can increase the firm's debt capacity because imperfectly correlated cash flow reduces the firm's overall risk. Kuppuswamy and Villalonga (2015) show that during the financial crisis period, diversified firms have better access to credit markets than single-segment firms (more money effect), as a result of the debt coinsurance provided by conglomerates. Since the insurance industry is a highly regulated industry, regulators monitor the solvency of insurers. Insurers commonly have held much more capital than required by regulation (Harrington 2005; Klein and Wang 2009). Therefore, during a financial crisis period, insurers may benefit less from the debt coinsurance effect of diversification than other firms.

We hypothesize that, under extreme economic conditions, the effect of diversification on firm performance is also contingent on the insurer's ownership structure. The diversification effect in different organization forms should be different since the financial crisis exacerbates the agency problem and highlights the importance of accessing external capital.

The diversification benefit, increasing debt capacity, should be more pronounced for mutual firms than stock firms during the financial crisis period since mutual firms cannot access equity markets. However, we do not expect this effect to be very significant as insurance companies use a very limited amount of debt. Mutual insurers do not have access to equity markets under normal conditions, so the fact that external capital becomes more expensive during the financial crisis, should not make a difference. However, we do expect the diversification cost of inefficient internal market allocation due to agency problems, to be more severe in mutual firms than in stock firms since monitoring managers is more difficult in mutual firms than in stock firms. During the financial crisis, the performance in the insurance industry fell significantly. Appendix B and Figure 4 show that profits after taxes fell to $3B (in 2016 dollars) in 2008 compared to a median profit of $40B during the 20 years between 1997 and 2016[2]. Diversified firms typically allocate resources to their best use (Rajan et al. (2000),

---

2   Figures 1 and 5 show that the reduction in profits was due to a reduction in net premiums written and a reduction in investment income.



Stein (1997), Weston (1970), Williamson (1975)) and diversification allows firms to form an internal capital market where the internally generated cash flows can be pooled. However, when resources and investment opportunities become extremely scarce (as they did during the financial crisis), we expect higher agency problems in diversified mutual firms as they are more difficult to monitor. Consequently, we examine the diversification effect during the financial crisis period in mutual firms and stock firms separately and we expect a reduction in the diversification premium or increase in the diversification penalty for mutual insurers. Our first set of competing hypothesis can be formally stated as:

**Hypothesis 1a.** *The effect of diversification for mutual insurers is negatively impacted by the financial crisis.*

**Hypothesis 1b.** *The effect of diversification for mutual insurers is positively by the financial crisis.*

We also study the effect of diversification during the financial crisis for groups and non-group insurers, separately. Given that group-affiliated insurers have access to larger internal capital markets through their groups, the presence of external financial constraints during the financial crisis should not significantly affect the diversification effect for group insurers. For non-group insurers, on the other hand, the relative value of an internal capital market provided by diversification under normal conditions is higher but when internal resources become extremely constrained, this benefit is no longer valuable. Thus, we expect a reduction in the diversification premium or increase in the diversification penalty for non-group insurers. Our second set of competing hypothesis can be formally stated as:

**Hypothesis 2a.** *The effect of diversification for non-group insurers is negatively impacted by the financial crisis.*

**Hypothesis 2b.** *The effects of diversification for non-group insurers is positively impacted by the financial crisis.*

## 4. Data and Methodology

### 4.1. Data

Our initial sample includes all firms in the National Association of Insurance Commissioners (NAIC) database for the years 2004–2013. We first exclude unrealistic numbers, such as negative total admitted asset and negative policyholder surplus (the same as equity in corporate firms). Then, we exclude the insurers that have negative total net premium written, which means they are not active firms in P/L insurance. We then aggregate affiliated insurers since the diversification decision is a group level decision. We also exclude insurers with substantial premium income (at least 25 percent of total premiums) from L/H insurance since our focus is on P/L insurers. Since our sample clearly has outliers, we exclude the 1st percentile and the 99th percentile of ROA and ROE. Finally, we exclude firms with organizational structures other than stock, mutual, group and non-group. Thus, our full sample has a total of 7470 firm-year observations.

We estimate a series of models to test the following relation:

$$Performance = f\ (diversification,\ economic\ activity,\ ownership\ structure,\ controls)$$

### 4.2. Performance Measures

Return on assets (ROA) and return on equity (ROE) are commonly used as accounting performance measurements. These accounting performance measures are also used in most of the insurance diversification literature (e.g., Hamilton and Shergill 1993; Mayer and Whittington 2003; Elango et al. 2008; Liebenberg and Sommer 2008). Following prior insurance diversification research, we use ROA and ROE as proxies of an insurer's accounting performance.

### 4.3. Diversification Measure

The diversification performance literature focusing on the insurance industry uses diversification measures that are binary, discrete or continuous. Liebenberg and Sommer (2008) use a discrete measure to distinguish between undiversified firms operating in only one business line and diversified firms operating in multiple business lines. Elango et al. (2008) use business Herfindahl indices to capture the magnitude of diversification. We use a binary measure of diversification following Liebenberg and Sommer (2008).[3] In the general finance literature, a binary measure of diversification is also the most commonly used (e.g., Berger and Ofek (1995), Campa and Kedia (2002)).

### 4.4. Economic Activity Measure

Our objective is to study the diversification effect for firms with different ownership structures and under different economic conditions. Specifically, we test whether the impact of diversification on firm performance is affected by the presence of external financial constraints. We choose to focus on the most recent financial crisis because of its magnitude and because its external and mostly unpredicted nature makes it an ideal setting for studying the effects of corporate finance (Campello et al. (2008, 2011)).

Kuppuswamy and Villalonga (2015) study the diversification effects during the financial crisis and they focus on all firms in COMPUSTAT except for the financial sector. They divide their sample period into four distinct subperiods: Pre-Crisis (2005Q1–2007Q2), Early Crisis (2007Q3–2008Q3), Late Crisis (2008Q4–2009Q1), and Post-Crisis (2009Q2–2009Q4). However, 2008Q4 and 2009Q1 are the peak time of the financial crisis. By then, the financial crisis had spilled over to the demand side (Ivashina and Scharfstein (2010), Kahle and Stulz (2010)). The insurance industry, as part of the financial sector, is considered to be on the demand side of the market (Gorton 2009). Since we only have yearly insurance statutory data, we divide our sample into two distinct sub-periods: Crisis (2008–2009) and Non-crisis (or Normal economic period from 2004–2007 and 2010–2013).

### 4.5. Ownership Structure

*Mutual* vs. *Stock.* As we discussed above, mutual firms and stock firms have different inherent costs and benefits. Previous studies have shown that mutual insurers underperform stock insurers. We use a dummy variable, MUTUAL, equal to 1 if an insurer's ownership structure is mutual, and 0 otherwise. We expect a negative relation between MUTUAL and performance.

*Group* vs. *Non-group.* Previous literature often finds that insurers in a group underperform unaffiliated insurers. Cummins and Sommer (1996) and Sommer (1996) argue that an insurance group has the option to let one of its affiliates fail to protect its other assets. Colquitt and Sommer (2003) find that group insurers with more complex structures require higher operating costs that offset any group economies. Thus, policyholders may consider group insurers riskier then unaffiliated insurers and be willing to pay more for the unaffiliated insurer's product. We use a dummy variable, GROUP, equal to one if the observation is an affiliated insurer, and 0 otherwise. We expect a negative relation between group status and firm performance.

### 4.6. Control Variables

*Firm Size.* In a very recent study, Dang et al. (2018) suggest that the results of some previous studies in the area of corporate finance are not robust to different proxies of firm size. They find that the regressions R-squared vary significantly with the choice of proxy for firm size (total assets, total sales, market capitalization), implying that some measures may be more appropriate than others in certain circumstances. They conclude that different proxies measure different aspects of firm size and

---

[3] As a robustness test, we use two additional measures of diversification, the number of lines and a modified business Herfindahl index. In our sample, we have a total of 23 business lines (see Appendix C for the business lines detail), which are the same as those in Liebenberg and Sommer (2008).

have different implications. Therefore, the choice of proxy should have a theoretical and empirical justification. The standard proxy for firm size in the insurance industry is the total admitted assets as other measures used in the general finance literature are not available or relevant. Prior literature finds that larger P/L insurers have higher returns on total admitted assets because of the presence of economies of scale, greater market power, and lower insolvency risk (Gardner and Grace 1993; Sommer 1996; Cummins and Nini 2002; Liebenberg and Sommer 2008). Therefore, we measure size as the natural logarithm of total admitted assets and expect a positive relationship between firm size and insurer's performance.

*Capitalization.* Sommer (1996) documents that insurers with lower insolvency risk are able to charge higher prices. Therefore, we expect that capitalization is positively related with performance. We measure capitalization as policyholder surplus divided by total admitted assets.

*Geographic Diversification.* In the insurance literature, geographic diversification is often used in regression models to control for its effects on firm performance (e.g., Mayers and Smith 1988, 1994; Pottier and Sommer 1997, McShane et al. 2010). Studies on insurance industry diversification also control for geographic diversification (e.g., Liebenberg and Sommer 2008; Elango et al. 2008). On one hand, geographically diversified firms require higher monitoring, and therefore they suffer higher monitoring costs, resulting in reduced financial performance (Mayers and Smith 1988). On the other hand, due to the coinsurance effect, geographically diversified firms are less likely to have volatile profits, suggesting they have lower risk. Geographically diversified firms are able to charge higher prices because of their lower risk compared with geographically focused firms. Thus, the relation between firm performance and geographic diversification is uncertain. Geographic diversification (GEODIV) is measured as one minus geographic Herfindahl, which is calculated as the Herfindahl index of direct premium written across all U.S. states and protectorates.

*Industry Concentration.* According to Montgomery (1985), firms that operate in more concentrated industries are more likely to benefit from higher prices, suggesting a positive relation between industry concentration and prices. A positive relation between market concentration and prices in P/L insurance lines is found by Chidambaran et al. (1997). In order to control for the impact of the competitiveness of firms' markets on performance, we follow Liebenberg and Sommer (2008). We calculate the weighted sum of market share per line multiplied by a line specific Herfindahl (WCONC). First, we calculate the Herfindahl index for each line of business (total 23 lines) across all firms in each year. Second, we calculate each firm's direct premiums written divided by the total direct premiums in each line of business in each year as the weight. Finally, we multiply the weight and Herfindahl index for each line of business across all firms in each year, and then sum the product for each firm by each year as each firm's yearly WCONC.

*Publicly Traded.* A publicly traded insurer may experience more monitoring from shareholders and more stringent scrutiny from analysts and regulators. Therefore, publicly traded insurers have a more effective market for corporate control than private insurers. We use a dummy variable, Public, to distinguish whether an insurer is publicly traded and we expect publicly traded insurers to outperform private insurers.

*Other Control Variables.* Although all the insurers in our sample write P/L insurance, several firms in the sample also write L/H business. We control for the percentage of premiums from life–health insurance (PCTLH). We also use the standard deviation of ROA over a 5-year period (SDROA5) as a measure of risk. We include year dummies to control for time-induced variation in performance, which is especially important for our sample given that it includes the financial crisis period. Finally, we also control for state and line dummies.

### 4.7. Methodology

In our multivariate analysis, we use an interaction between MULTILINE and CRISIS89 to estimate the combined effect of diversification and the financial crisis on firm performance for different ownership

structures. Our basic model measuring the effect of diversification on performance is defined in Equations (1) and (2).

$$
\begin{aligned}
ROA_{it} = {}& \beta_0 + \beta_1 MULTILINE_{it} + \beta_2 MULTILINE_{it} * CRISIS89_{it} + \beta_3 CRISIS89_{it} + \beta_4 SIZE_{it} \\
& + \beta_5 CAPASSET_{it} + \beta_6 WCONC_{it} + \beta_7 GEODIV_{it} + \beta_7 GROUP_{it} \\
& + \beta_8 PUBLIC_{it} + \beta_9 PCTLH_{it} + \beta_{10} SDROA5_{it} + YEAR\ dummies \\
& + STATE\ dummies + LINE\ dummies + \varepsilon_{it}
\end{aligned}
\tag{1}
$$

$$
\begin{aligned}
ROA_{it} = {}& \beta_0 + \beta_1 MULTILINE_{it} + \beta_2 MULTILINE_{it} * CRISIS89_{it} + \beta_3 CRISIS89_{it} + \beta_4 SIZE_{it} \\
& + \beta_5 CAPASSET_{it} + \beta_6 WCONC_{it} + \beta_7 GEODIV_{it} + \beta_7 MUTUAL_{it} \\
& + \beta_8 PUBLIC_{it} + \beta_9 PCTLH_{it} + \beta_{10} SDROA5_{it} + YEAR\ dummies \\
& + STATE\ dummies + LINE\ dummies + \varepsilon_{it}
\end{aligned}
\tag{2}
$$

Equation (1) tests H1 and we estimate it for the full sample and also for the mutual and stock sub-samples. Equation (2) tests H2 and we estimate it for the full sample and also for the group and non-group sub-samples.

Campa and Kedia (2002) and Villalonga (2004, 2015) have shown that studies of the effect of diversification on firm performance/value suffer from a self-selection problem. Firms self-select into or out of their diversification status. Self-selection is one of the three types of endogeneity (simultaneous causality, omitted variables, and measurement error). A common source of omitted variables bias is self-selection. We conduct a Hausman test to determine whether we have an endogeneity problem. First, we employ a logit regression on MULTILINE using other control variables as independent variables and get a residual. Then, we plug the residual into Equation (1). If the residual term in Equation (1) is significant, we have an endogeneity problem in the model. The Hausman test shows that the residual term is significant, suggesting the presence of endogeneity.

There are several corrections for endogeneity suggested in the literature. Li (2016) in his experimental research of endogeneity in CEO power, finds that all the prevailing econometric methods are generally effective in mitigating the endogeneity problem to a certain degree. Researchers that have studied the endogeneity problem in the diversification discount literature have offered three solutions. The first technique is a fixed effects regression model in which one can control for unobserved or omitted firm-specific effects that may be correlated with other independent variables. However, Wooldridge (2002) shows that the effectiveness of the fixed effects model is limited to situations in which there is sufficient within-firm variation. Liebenberg and Sommer (2008) argue that a fixed effects model would be inadequate in their study given that there is very little variation in their MULTILINE variable. Most firms seem to keep their diversification status over time. Our sample suffers from the same problem. Thus, we are not able to use firm fixed effects but we do estimate Equations (1) and (2) using an OLS model with year, state, and lines fixed effects.

The other two techniques to correct for endogeneity are two-stage least squares (2SLS) and a Heckman (treatment effects) model. In the general finance literature, Campa and Kedia (2002), Villalonga (2004), and Laeven and Levine (2007) have used these models. McCullough and Hoyt (2005) and Liebenberg and Sommer (2008) have used these techniques in the diversification discount literature within the insurance industry.

We use both models, a 2SLS and a Heckman model, to correct for endogeneity. The two approaches require a selection of instrumental variables. Successful instrumental variable candidates must satisfy two criteria. The first criterion is that the instruments correlate with MULTILINE. The second criterion is that the instruments do not correlate with the error term in Equation (1). Following Liebenberg and Sommer (2008), we choose reinsurance use and an index reflecting the attractiveness of a firm's markets to single-line insurers as our instrument variables. The Hansen's J-test of overidentifying restrictions is not significant, indicating that the instruments are uncorrelated with the error term in Equation (1), and the two instruments are valid. We repeat the same procedure with Equation (2).

Finally, Bascle (2008) argues that the choice between these two models should be based on the type of endogeneity with which the researcher is confronted. Wooldridge (2002) and Heckman (1979), among several others, contend that a Heckman two-step model is the solution for self-selection. We will present results with these three corrections for endogeneity, but our primary focus should be the Heckman two-step model.

## 5. Results

### 5.1. Summary Statistics

Table 1 provides variable definition and descriptive statistics for our sample and compare them with the sample in Liebenberg and Sommer (2008). In our sample, we have 7470 firm-year observations from 2004 to 2013. Liebenberg and Sommer's (2008) used 6290 firm-year observations from 1995 to 2004. All the variables statistics are comparable, except LINES and MULTILINE. The mean and median of the number of lines are 4.49 and 3.00, which are smaller than 5.91 and 5.00 in the Liebenberg and Sommer (2008) sample. Fewer firms in our sample are diversified insurers than in the Liebenberg and Sommer (2008) sample. One possible explanation is that diversified insurers with poor performance during 1995–2004 decided to refocus or cut lines.

Table 2 shows the descriptive statistics comparison of mutual firms and stock firms under the crisis and non-crisis periods in Panel A and the comparison for group insurers vs. non-groups in Panel B. As expected, the performance of all groups during the crisis period is lower than the performance during the normal period. In Panel A, the average and median performance of mutual insurers is similar to that of stock insurers. The average and median number of lines in the mutual sample are greater than in the stock sample. By comparing MULTILINE and the modified business Herfindahl, mutual insurers are more diversified than stock insurers. In Panel B, the average and median performance of group insurers is similar to that of non-group insurers. The average and median number of lines in the group sample are greater than in the non-group sample. By comparing MULTILINE and the modified business Herfindahl, group insurers are more diversified than non-group insurers.

### 5.2. Univariate Results

In this section, we compare the mean and median performance of diversified insurers and single-line for each of our sub-samples to determine whether there is a diversification premium or penalty. Panel A of Table 3 shows that under normal conditions, there is a diversification premium for mutual insurers but it turns into a diversification penalty during the financial crisis, supporting H1a. We find similar results for non-group insurers in Panel D, diversification improves the performance of non-group insurers under normal conditions, but it reduces their performance during the financial crisis, supporting H2a. The results for stock insurers and group insurers are not as striking. Panel B shows that for stock insurers, a small premium under normal conditions disappears during the financial crisis. Panel C shows that the performance of diversified group insurers is not statistically different from single-line groups regardless of the economic conditions.

**Table 1.** Variable definition and descriptive statistics compared with that in Liebenberg and Sommer (2008).

| Variables | Definition | Full Sample (7470 Firm-Years) | | | Liebenberg and Sommer (2008) (6290 Firm-Years) | | |
|---|---|---|---|---|---|---|---|
| | | Mean | Median | Standard Deviation | Mean | Median | Standard Deviation |
| ROA | Net income/total admitted assets | 0.02 | 0.02 | 0.06 | 0.02 | 0.03 | 0.05 |
| ROE | Net income/policyholder surplus | 0.03 | 0.05 | 0.14 | 0.05 | 0.06 | 0.14 |
| LINES | Number of lines in which firm has positive direct premiums written (DPW) | 4.49 | 3.00 | 3.97 | 5.91 | 5.00 | 4.63 |
| MULTILINE | = 1 if LINES > 1, 0 otherwise | 0.66 | 1.00 | 0.48 | 0.79 | 1.00 | 0.41 |
| SIZE | Natural Logarithm of total admitted assets | 17.83 | 17.57 | 2.28 | 17.64 | 17.47 | 2.19 |
| CAPASSET | Policyholder surplus/total admitted assets | 0.49 | 0.46 | 0.20 | 0.49 | 0.44 | 0.21 |
| GEODIV | 1-Herfindahl index of DPW across 57 geographic areas | 0.30 | 0.02 | 0.36 | 0.33 | 0.11 | 0.37 |
| WCONC | Weighted sum of market share per line multiplied by line specific Herfindahl | 0.05 | 0.05 | 0.01 | 0.05 | 0.05 | 0.02 |
| MUTUAL | = 1 if firm is a mutual, 0 otherwise | 0.45 | 0.00 | 0.50 | 0.48 | 0.00 | 0.05 |
| GROUP | = 1 if firm is a group, 0 otherwise | 0.25 | 0.00 | 0.43 | 0.34 | 0.00 | 0.47 |
| PCTLH | Percentage of premiums from life-health insurance | 0.23 | 0.00 | 7.76 | 0.44 | 0.00 | 2.16 |
| PUBLIC | = 1 if firm is publicly traded, 0 otherwise | 0.05 | 0.00 | 0.21 | 0.08 | 0.00 | 0.47 |
| MBH | 1-business Herfindahl | 0.31 | 0.27 | 0.31 | | | |

**Table 2.** Sample descriptive Statistics Comparison.

**Panel A: Mutual vs. stock**

| Variables | Definition | Mutual Sample (3355 Firm-Years) | | | | | | Stock Sample (4115 Firm-Years) | | | | | |
|---|---|---|---|---|---|---|---|---|---|---|---|---|---|
| | | Crisis | | | Non-crisis | | | Crisis | | | Non-Crisis | | |
| | | Mean | Median | Standard Deviation | Mean | Median | Standard Deviation | Mean | Median | Standard Deviation | Mean | Median | Standard Deviation |
| ROA | Net income/total admitted assets | 0.01 | 0.02 | 0.05 | 0.02 | 0.03 | 0.05 | 0.01 | 0.02 | 0.06 | 0.02 | 0.03 | 0.06 |
| ROE | Net income/policyholder surplus | 0.01 | 0.03 | 0.13 | 0.04 | 0.05 | 0.13 | 0.02 | 0.04 | 0.15 | 0.04 | 0.06 | 0.15 |
| LINES | Number of lines in which firm has positive direct premiums written (DPW) | 5.40 | 5.00 | 3.81 | 5.36 | 5.00 | 3.84 | 3.71 | 2.00 | 3.83 | 3.79 | 2.00 | 3.97 |
| MULTILINE | = 1 if LINES > 1, 0 otherwise | 0.76 | 1.00 | 0.43 | 0.76 | 1.00 | 0.43 | 0.58 | 1.00 | 0.49 | 0.57 | 1.00 | 0.50 |
| SIZE | Natural Logarithm of total admitted assets | 17.91 | 17.93 | 2.38 | 17.90 | 17.92 | 2.40 | 17.72 | 17.39 | 2.14 | 17.78 | 17.40 | 2.19 |
| CAPASSET | Policyholder surplus/total admitted assets | 0.51 | 0.49 | 0.20 | 0.51 | 0.48 | 0.20 | 0.49 | 0.44 | 0.21 | 0.47 | 0.43 | 0.20 |
| GEODIV | 1-Herfindahl index of DPW across 57 geographic areas | 0.24 | 0.00 | 0.34 | 0.25 | 0.00 | 0.34 | 0.34 | 0.18 | 0.37 | 0.35 | 0.18 | 0.37 |
| WCONC | Weighted sum of market share per line multiplied by line specific Herfindahl | 0.05 | 0.05 | 0.01 | 0.05 | 0.05 | 0.01 | 0.06 | 0.06 | 0.01 | 0.05 | 0.05 | 0.02 |
| GROUP | = 1 if firm is a group, 0 otherwise | 0.33 | 0.00 | 0.47 | 0.33 | 0.00 | 0.47 | 0.18 | 0.00 | 0.38 | 0.19 | 0.00 | 0.39 |
| PCTLH | Percentage of premiums from life-health insurance | 0.41 | 0.00 | 2.23 | 0.47 | 0.00 | 2.57 | 0.22 | 0.00 | 1.86 | 0.01 | 0.00 | 11.45 |
| PUBLIC | = 1 if firm is publicly traded, 0 otherwise | 0.02 | 0.00 | 0.14 | 0.02 | 0.00 | 0.14 | 0.07 | 0.00 | 0.25 | 0.07 | 0.00 | 0.25 |
| MBH | 1-business Herfindahl | 0.42 | 0.54 | 0.30 | 0.41 | 0.54 | 0.30 | 0.22 | 0.03 | 0.32 | 0.23 | 0.03 | 0.28 |

**Panel B: Group vs. non-group**

| Variables | Definition | Group Sample (1876 Firm-Years) | | | | | | Non-Group Sample (5594 Firm-Years) | | | | | |
|---|---|---|---|---|---|---|---|---|---|---|---|---|---|
| | | Crisis | | | Non-crisis | | | Crisis | | | Non-crisis | | |
| | | Mean | Median | Standard Deviation | Mean | Median | Standard Deviation | Mean | Median | Standard Deviation | Mean | Median | Standard Deviation |
| ROA | Net income/total admitted assets | 0.01 | 0.02 | 0.04 | 0.03 | 0.03 | 0.04 | 0.01 | 0.02 | 0.06 | 0.02 | 0.03 | 0.06 |
| ROE | Net income/policyholder surplus | 0.03 | 0.05 | 0.09 | 0.06 | 0.06 | 0.10 | 0.01 | 0.03 | 0.15 | 0.03 | 0.05 | 0.15 |
| LINES | Number of lines in which firm has positive direct premiums written (DPW) | 8.03 | 8.00 | 4.60 | 8.16 | 8.00 | 4.68 | 3.30 | 2.00 | 2.82 | 3.26 | 2.00 | 2.80 |
| MULTILINE | = 1 if LINES > 1, 0 otherwise | 0.91 | 1.00 | 0.29 | 0.92 | 1.00 | 0.28 | 0.58 | 1.00 | 0.49 | 0.57 | 1.00 | 0.50 |
| SIZE | Natural Logarithm of total admitted assets | 20.46 | 20.32 | 1.65 | 20.47 | 20.35 | 1.66 | 16.95 | 16.84 | 1.68 | 16.94 | 16.84 | 1.71 |
| CAPASSET | Policyholder surplus/total admitted assets | 0.44 | 0.41 | 0.13 | 0.43 | 0.41 | 0.13 | 0.52 | 0.49 | 0.22 | 0.51 | 0.48 | 0.22 |
| GEODIV | 1-Herfindahl index of DPW across 57 geographic areas | 0.58 | 0.70 | 0.35 | 0.58 | 0.71 | 0.35 | 0.21 | 0.00 | 0.32 | 0.21 | 0.00 | 0.32 |
| WCONC | Weighted sum of market share per line multiplied by line specific Herfindahl | 0.05 | 0.05 | 0.01 | 0.05 | 0.05 | 0.01 | 0.05 | 0.05 | 0.01 | 0.05 | 0.05 | 0.02 |
| MUTUAL | = 1 if firm is a mutual, 0 otherwise | 0.59 | 1.00 | 0.49 | 0.58 | 1.00 | 0.49 | 0.39 | 0.00 | 0.49 | 0.41 | 0.00 | 0.49 |
| PUBLIC | = 1 if firm is publicly traded, 0 otherwise | 0.17 | 0.00 | 0.37 | 0.17 | 0.00 | 0.38 | 0.01 | 0.00 | 0.08 | 0.01 | 0.00 | 0.07 |
| MBH | 1-business Herfindahl | 0.47 | 0.57 | 0.37 | 0.49 | 0.58 | 0.28 | 0.25 | 0.04 | 0.29 | 0.25 | 0.04 | 0.29 |

**Table 3.** Univariate comparison between diversified and single-line insurers during crisis and non-crisis period.

**Panel A: Mutual**

| Variable | Crisis | | | | | | Non-crisis | | | | | |
|---|---|---|---|---|---|---|---|---|---|---|---|---|
| | Diversified Insurers | | Single-Line Insurers | | Differences Tests | | Diversified Insurers | | Single-Line Insurers | | Differences Tests | |
| | Mean | Median | Mean | Median | Mean | Median | Mean | Median | Mean | Median | Mean | Median |
| ROA | 0.005 | 0.016 | 0.016 | 0.017 | −0.011 ** | −0.001 | 0.023 | 0.029 | 0.016 | 0.021 | 0.007 *** | 0.008 *** |
| ROE | 0.003 | 0.032 | 0.046 | 0.038 | −0.043 *** | −0.006 *** | 0.043 | 0.057 | 0.028 | 0.041 | 0.015 *** | 0.015 *** |

**Panel B: Stock**

| Variable | Crisis | | | | | | Non-crisis | | | | | |
|---|---|---|---|---|---|---|---|---|---|---|---|---|
| | Diversified Insurers | | Single-Line Insurers | | Differences Tests | | Diversified Insurers | | Single-Line Insurers | | Differences Tests | |
| | Mean | Median | Mean | Median | Mean | Median | Mean | Median | Mean | Median | Mean | Median |
| ROA | 0.012 | 0.019 | 0.011 | 0.015 | 0.001 | 0.004 | 0.021 | 0.027 | 0.018 | 0.023 | 0.003 | 0.004 * |
| ROE | 0.022 | 0.045 | 0.010 | 0.032 | 0.012 | 0.014 | 0.045 | 0.063 | 0.033 | 0.051 | 0.012 ** | 0.011 *** |

**Panel C: Group**

| Variable | Crisis | | | | | | Non-crisis | | | | | |
|---|---|---|---|---|---|---|---|---|---|---|---|---|
| | Diversified Insurers | | Single-Line Insurers | | Differences Tests | | Diversified Insurers | | Single-Line Insurers | | Differences Tests | |
| | Mean | Median | Mean | Median | Mean | Median | Mean | Median | Mean | Median | Mean | Median |
| ROA | 0.013 | 0.020 | 0.016 | 0.023 | −0.003 | −0.003 | 0.025 | 0.028 | 0.027 | 0.024 | −0.002 | 0.004 |
| ROE | 0.029 | 0.045 | 0.049 | 0.072 | −0.020 | −0.027 | 0.056 | 0.064 | 0.077 | 0.064 | −0.022 ** | 0.000 |

**Panel D: Non-group**

| Variable | Crisis | | | | | | Non-crisis | | | | | |
|---|---|---|---|---|---|---|---|---|---|---|---|---|
| | Diversified Insurers | | Single-Line Insurers | | Differences Tests | | Diversified Insurers | | Single-Line Insurers | | Differences Tests | |
| | Mean | Median | Mean | Median | Mean | Median | Mean | Median | Mean | Median | Mean | Median |
| ROA | 0.006 | 0.017 | 0.012 | 0.016 | −0.006 * | 0.001 | 0.020 | 0.027 | 0.017 | 0.021 | 0.003 * | 0.006 *** |
| ROE | 0.004 | 0.032 | 0.019 | 0.033 | −0.015 * | −0.001 | 0.037 | 0.055 | 0.029 | 0.046 | 0.009 * | 0.009 ** |

Single-line insurers are those firms that only have one business line (MULTILNE = 0). Diversified insurers are those firms that have more than one business line (MULTILINE = 1). Return on assets (ROA) is net income divided by total admitted assets. Return on equity (ROE) is net income divided by policyholder surplus. Crisis period is 2008 and 2009. The difference of means is tested with a *t*-test and the difference of medians is tested with a Wilcoxon rank sum test. Statistical significance at the 1%, 5% and 10% levels is denoted by ***, **, *, respectively.

*5.3. Regression Analysis*

Table 4 shows the effect of diversification on ROA using OLS, 2SLS, and the Heckman model, for mutual and stock insurers. Our regression analysis confirms our univariate results. The OLS and Heckman models show that the performance of diversified mutual insurers is significantly reduced during the financial crisis relative to the performance of single-line mutual insurers. This finding is consistent with our hypothesis H1a that the scarcity of internal resources and lack of investment opportunities during the financial crisis increases the costs of diversification (agency problems) for mutual insurers as they are less monitored than stock insurers. In contrast, the diversification effect for stock insurers is not significantly affected by the extreme economic conditions of the financial crisis.

Table 5 examines the diversification effect on ROA for group vs. non-group insurers. The results show that our univariate results are robust to the inclusion of control variables and a correction for endogeneity. The performance of diversified non-group insurers is significantly reduced during the financial crisis relative to single-line insurers. This result implies that the diversification premium found in the univariate analysis under normal conditions is reversed to a diversification discount under extreme economic conditions. Our finding for non-group insurers is consistent with our hypothesis H2a that the increased the relative value of an internal capital market provided by diversification under normal conditions is higher for non-groups than for group insurers, but when internal resources become extremely constrained, this benefit is no longer valuable. Once this benefit is eliminated, the costs must become larger than the benefits, causing a diversification penalty.

*5.4. Robustness Tests*[4]

5.4.1. Alternative Measure of Performance: ROE

To investigate whether our results are robust to an alternative measure of performance, we replicate Tables 4 and 5 using ROE as the dependent variable instead of ROA. We test hypotheses H1 and H2 using an OLS model with year, state, and line fixed effects and find consistent results with those reported in Tables 4 and 5. The performance of diversified mutual and non-group insurers is significantly reduced during the financial crisis while the performance of diversified stock and group insurers is not significantly affected by the extreme economic conditions that we experienced during the Great Recession.

5.4.2. Alternative Measure of Diversification: Business Herfindahl

The diversification literature in the insurance industry also uses business Herfindahl indexes to measure diversification (e.g., Elango et al. 2008). As a robustness test, we examine the diversification effects using a continuous measure of diversification, business Herfindahl. In order to be consistent with the interpretation of the MULTILINE variable, we use 1 minus business Herfindahl to capture the extent of diversification (MBH). Using ROA as the dependent variable and an OLS model with year, state, and line fixed effects, we find that the performance of diversified mutual and non-group insurers is significantly reduced during the financial crisis. These results are consistent with our main results obtained when using a binary measure of diversification.

5.4.3. Alternative Measure of Diversification: Number of Lines

We also use number of lines (LINES) as an alternative measure of diversification (instead of our binary variable, MULTILINE). We test H1 and H2 using ROA as our dependent variable and LINES as our key independent variable. Our results are consistent with our findings in Tables 4 and 5.

---

[4]  These additional results are available from the authors upon request.

**Table 4.** Diversification Effect on Performance (ROA), Mutual vs. Stock.

| | Dependent Variable: ROA | | | | | | | | |
|---|---|---|---|---|---|---|---|---|---|
| **Model** | **Full Sample** | | | **Mutual Sample** | | | **Stock Sample** | | |
| | **OLS** | **2SLS** | **HECKMAN** | **OLS** | **2SLS** | **HECKMAN** | **OLS** | **2SLS** | **HECKMAN** |
| MULTILINE | −0.00372 | −0.0564 | −0.0324 *** | −0.00179 | −0.0930 | −0.0145 | −0.00499 | 0.128 | −0.0144 |
| | (0.00357) | (0.0545) | (0.00600) | (0.00552) | (0.0591) | (0.00964) | (0.00447) | (0.106) | (0.00878) |
| MULTILINE * Crisis89 | −0.00634 * | 0.0147 | −0.00621 ** | −0.0111 *** | 0.0157 | −0.0108 ** | −0.00227 | −0.0575 | −0.00232 |
| | (0.00330) | (0.0219) | (0.00309) | (0.00413) | (0.0176) | (0.00445) | (0.00457) | (0.0444) | (0.00430) |
| Crisis89 | 0.0168 *** | −0.0140 | −0.000734 | 0.000182 | −0.0179 | 0.00253 | −0.00235 | 0.0279 | −0.00239 |
| | (0.00347) | (0.0141) | (0.00335) | (0.00449) | (0.0144) | (0.00478) | (0.00469) | (0.0247) | (0.00463) |
| size | 0.00794 *** | 0.00863 *** | 0.00782 *** | 0.0106 *** | 0.0105 *** | 0.0105 *** | 0.00576 *** | 0.00183 | 0.00589 *** |
| | (0.00103) | (0.00133) | (0.000519) | (0.00121) | (0.00139) | (0.000723) | (0.00153) | (0.00365) | (0.000824) |
| capasset | 0.0595 *** | 0.0559 *** | 0.0639 *** | 0.0687 *** | 0.0698 *** | 0.0706 *** | 0.0551 *** | 0.0638 *** | 0.0558 *** |
| | (0.00698) | (0.00797) | (0.00398) | (0.0101) | (0.0117) | (0.00598) | (0.00986) | (0.0140) | (0.00575) |
| wconc | −0.0874 | 0.0427 | −0.0926 | −0.209 | 0.325 | −0.199 ** | −0.0724 | −0.227 | −0.0706 |
| | (0.0921) | (0.167) | (0.0586) | (0.153) | (0.403) | (0.0970) | (0.124) | (0.203) | (0.0819) |
| geodiv | −0.00650 | −0.00527 | −0.00544 | 0.00328 | 0.0113 | 0.00355 | −0.0168 ** | −0.0131 | −0.0164 *** |
| | (0.00531) | (0.00596) | (0.00345) | (0.00650) | (0.0104) | (0.00556) | (0.00764) | (0.0109) | (0.00512) |
| group | −0.0180 *** | −0.0209 *** | −0.0192 *** | −0.0159 *** | −0.0155 *** | −0.0164 *** | −0.0182 *** | −0.00633 | −0.0185 *** |
| | (0.00316) | (0.00480) | (0.00223) | (0.00355) | (0.00443) | (0.00286) | (0.00528) | (0.0130) | (0.00382) |
| public | −0.00314 | −0.00683 | −0.00317 | −0.00517 | −0.0176 | −0.00504 | −0.00354 | 0.00719 | −0.00390 |
| | (0.00393) | (0.00596) | (0.00347) | (0.00643) | (0.0126) | (0.00719) | (0.00497) | (0.0112) | (0.00446) |
| percentagelh | 0.00245 | 0.00237 | 0.00262 | −0.0295 | −0.0675 | −0.0271 | 0.00840 *** | 0.00385 | 0.00850 |
| | (0.00496) | (0.00690) | (0.00760) | (0.0492) | (0.0646) | (0.0412) | (0.00243) | (0.00531) | (0.00840) |
| sdroa5 | −0.197 *** | −0.208 *** | −0.203 *** | −0.0871 | −0.103 | −0.0875 *** | −0.294 *** | −0.282 *** | −0.296 *** |
| | (0.0592) | (0.0609) | (0.0160) | (0.0638) | (0.0644) | (0.0200) | (0.0561) | (0.0626) | (0.0252) |
| Constant | −0.152 *** | −0.143 *** | −0.118 *** | −0.189 *** | −0.198 *** | −0.180 *** | −0.0903 *** | −0.0392 | −0.0878 *** |
| | (0.0194) | (0.0220) | (0.0107) | (0.0238) | (0.0278) | (0.0155) | (0.0278) | (0.0526) | (0.0161) |
| Year FE | Yes | Yes | Yes | Yes | Yes | Yes | Yes | Yes | Yes |
| State FE | Yes | Yes | Yes | Yes | Yes | Yes | Yes | Yes | Yes |
| Line FE | Yes | Yes | Yes | Yes | Yes | Yes | Yes | Yes | Yes |
| Underidentification test | | 7.293 * | | | 6.541 * | | | 3.772 | |
| Overidentification test | | 9.545 *** | | | 0.288 | | | 2.509 | |

**Table 4.** *Cont.*

| Model | Dependent Variable: ROA | | | | | | | | |
|---|---|---|---|---|---|---|---|---|---|
| | Full Sample | | | Mutual Sample | | | Stock Sample | | |
| | OLS | 2SLS | HECKMAN | OLS | 2SLS | HECKMAN | OLS | 2SLS | HECKMAN |
| Self-selection parameter | | | 0.0174 *** | | | 0.00721 | | | 0.00580 |
| | | | (0.00334) | | | (0.00503) | | | (0.00502) |
| Number of observations | 6513 | 6513 | 6513 | 2970 | 2970 | 2970 | 3543 | 3543 | 3543 |
| Peseudo R square | 0.185 | 0.123 | | 0.264 | 0.119 | | 0.192 | −0.192 | |

Notes: The dependent variable is return on assets (ROA). OLS is an ordinary least squares model. 2SLS is a two-stage least square model. The first-stage regression is a logistic regression of MULTILINE on the other variables in Equation (1). The second-stage regression is an ordinary least squares regression of MULTILINE on the predicted value from the first-stage regression and instrument variables (reinsurance ratio and an index that captures the attractiveness of a firm's markets to single-line insurers). HECKMAN is a two-step treatment effects regression. MULTILINE is equal to one for insurers with more than one business line, and zero otherwise. Crisis89 is a dummy equal to 1 for years 2008 and 2009, and 0 otherwise. SIZE is equal to the natural logarithm of total admitted assets. CAPASSET is the ratio of policyholder surplus to total admitted assets. GROUP is equal to one for aggregated groups, zero otherwise. GEODIV is equal to one minus the Herfindahl index of premiums across 57 geographic areas. WCONC is the weighted sum of insurer market share per line multiplied by each line's Herfindahl index. PCTLH is the percentage of L/H premium over total premium (L/H and P/L). PUBLIC is equal to one if the insurer is publicly traded, zero otherwise. SDROA5 is the standard deviation of ROA over a 5-year period. Standard errors are in parentheses and they are adjusted for firm-level clustering. Statistical significance at the 1%, 5%, and 10% levels is denoted by ***, **, and *, respectively.

**Table 5.** Diversification Effect on Performance (ROA), Group vs. Non-Group.

| Model | Dependent Variable: ROA | | | | | | | | |
|---|---|---|---|---|---|---|---|---|---|
| | Full Sample | | | Group Sample | | | Non-Group Sample | | |
| | OLS | 2SLS | HECKMAN | OLS | 2SLS | HECKMAN | OLS | 2SLS | HECKMAN |
| MULTILINE | −0.00315 | −0.000449 | −0.0223 *** | −0.00119 | −0.0471 | −0.0249 *** | −0.00874 ** | −0.238 | −0.0303 *** |
| | (0.00359) | (0.0362) | (0.00607) | (0.00755) | (0.0617) | (0.00963) | (0.00406) | (0.214) | (0.00788) |
| MULTILINE * Crisis89 | −0.00624 * | −0.00732 | −0.00616 ** | −0.00109 | 0.0205 | 0.000244 | −0.00803 ** | 0.0754 | −0.00796 ** |
| | (0.00331) | (0.0148) | (0.00311) | (0.00590) | (0.0303) | (0.00588) | (0.00381) | (0.0778) | (0.00373) |
| Crisis89 | 0.0165 *** | −0.000133 | −0.00100 | 0.000998 | −0.0164 | 0.00164 | 0.0140 *** | −0.0466 | −0.00195 |
| | (0.00347) | (0.00975) | (0.00337) | (0.00637) | (0.0281) | (0.00622) | (0.00391) | (0.0419) | (0.00397) |
| size | 0.00638 *** | 0.00635 *** | 0.00617 *** | 0.00545 *** | 0.00511 *** | 0.00523 *** | 0.00957 *** | 0.0139 *** | 0.00947 *** |
| | (0.000885) | (0.000933) | (0.000477) | (0.00132) | (0.00135) | (0.000793) | (0.00137) | (0.00451) | (0.000661) |
| capasset | 0.0575 *** | 0.0577 *** | 0.0602 *** | 0.0755 *** | 0.0762 *** | 0.0799 *** | 0.0605 *** | 0.0583 *** | 0.0632 *** |
| | (0.00684) | (0.00727) | (0.00397) | (0.0133) | (0.0129) | (0.00747) | (0.00822) | (0.0142) | (0.00478) |
| wconc | −0.113 | −0.119 | −0.111 * | −0.426 *** | −0.343 * | −0.394 *** | −0.0594 | 0.251 | −0.0696 |

**Table 5.** *Cont.*

| Model | Dependent Variable: ROA | | | | | | | | |
| | Full Sample | | | Group Sample | | | Non-Group Sample | | |
| | OLS | 2SLS | HECKMAN | OLS | 2SLS | HECKMAN | OLS | 2SLS | HECKMAN |
|---|---|---|---|---|---|---|---|---|---|
| | (0.0936) | (0.127) | (0.0593) | (0.157) | (0.195) | (0.0903) | (0.107) | (0.365) | (0.0748) |
| geodiv | −0.00794 | −0.00799 | −0.00716 ** | −0.00705 | −0.00619 | −0.00755 * | −0.0169 ** | −0.0210 | −0.0160 *** |
| | (0.00532) | (0.00527) | (0.00347) | (0.00667) | (0.00692) | (0.00423) | (0.00733) | (0.0154) | (0.00492) |
| gmutual | −0.00717 *** | −0.00707 ** | −0.00584 *** | −0.00835 *** | −0.0103 ** | −0.00814 *** | −0.00449 | −0.0160 | −0.00220 |
| | (0.00255) | (0.00303) | (0.00169) | (0.00294) | (0.00410) | (0.00214) | (0.00323) | (0.0128) | (0.00229) |
| public | −0.00709 * | −0.00686 | −0.00700 ** | 0.00308 | 0.00161 | 0.00340 | −0.00108 | 0.0368 | −0.00127 |
| | (0.00395) | (0.00490) | (0.00349) | (0.00398) | (0.00425) | (0.00255) | (0.00965) | (0.0496) | (0.0110) |
| sdroa5 | −0.201 *** | −0.200 *** | −0.204 *** | −0.531 *** | −0.513 *** | −0.534 *** | −0.171 *** | −0.191 *** | −0.175 *** |
| | (0.0601) | (0.0611) | (0.0161) | (0.130) | (0.138) | (0.0476) | (0.0580) | (0.0622) | (0.0181) |
| Constant | −0.120 *** | −0.103 *** | −0.0900 *** | −0.0879 *** | −0.0565 | −0.0590 *** | −0.182 *** | −0.233 *** | −0.156 *** |
| | (0.0169) | (0.0167) | (0.0102) | (0.0274) | (0.0385) | (0.0184) | (0.0251) | (0.0732) | (0.0136) |
| Year FE | Yes | Yes | Yes | Yes | Yes | Yes | Yes | Yes | Yes |
| State FE | Yes | Yes | Yes | Yes | Yes | Yes | Yes | Yes | Yes |
| Line FE | Yes | Yes | Yes | Yes | Yes | Yes | Yes | Yes | Yes |
| Underidentification test | | 13.664 *** | | | 3.594 | | | 1.689 | |
| Overidentification test | | 6.524 *** | | | 4.649 * | | | 0.153 | |
| Self−selection parameter | | | 0.0117 *** | | | 0.0127 *** | | | 0.0131 *** |
| | | | (0.00342) | | | (0.00461) | | | (0.00444) |
| Number of observations | 6513 | 6513 | 6513 | 1668 | 1668 | 1668 | 4845 | 4845 | 4845 |
| Peseudo R square | 0.180 | 0.179 | | 0.392 | 0.352 | | 0.190 | −0.815 | |

Notes: The dependent variable is return on assets (ROA). OLS is an ordinary least squares model. 2SLS is a two-stage least square model. The first-stage regression is a logistic regression of MULTILINE on the other variables in Equation (1). The second-stage regression is an ordinary least squares regression of MULTILINE on the predicted value from the first-stage regression and instrument variables (reinsurance ratio and an index that captures the attractiveness of a firm's markets to single-line insurers). HECKMAN is a two-step treatment effects regression. MULTILINE is equal to one for insurers with more than one business line, and zero otherwise. Crisis89 is a dummy equal to 1 for years 2008 and 2009, and 0 otherwise. SIZE is equal to the natural logarithm of total admitted assets. CAPASSET is the ratio of policyholder surplus to total admitted assets. MUTUAL is equal to one for mutual insurers, zero otherwise. GEODIV is equal to one minus the Herfindahl index of premiums across 57 geographic areas. WCONC is the weighted sum of insurer market share per line multiplied by each line's Herfindahl index. PUBLIC is equal to one if the insurer is publicly traded, zero otherwise. SDROA5 is the standard deviation of ROA over a 5-year period. Standard errors are in parentheses and they are adjusted for firm-level clustering. Statistical significance at the 1%, 5%, and 10% levels is denoted by ***, **, and *, respectively.

## 6. Conclusions

Diversification effects have been studied extensively in recent years. The evidence of the financial crisis impacting diversification effects is documented in Kuppuswamy and Villalonga (2015) and Rudolph and Schwetzler (2013). However, in their research, only stock firms are studied, and the insurance industry is excluded from their sample. By focusing on the insurance industry, we benefit from the richness of the insurance statutory dataset and are able to explore the diversification effects on different ownership structures under different economic conditions. The Great Recession provides a natural environment for us to test the diversification effects among different ownership structures in the presence of external financing constraints and when the expectation of performance falls.

Using a sample of 7470 firm-year observations between 2004 and 2013, we find that the effect of diversification on performance is contingent on the different ownership structures of insurance firms and on different economic conditions. Mutual insurers experience a diversification premium under normal economic conditions but suffer a diversification penalty during the financial crisis. Monitoring managers in mutual firms is more difficult than in stock firms and monitoring managers in diversified firms is more difficult than in single-line firms. Our finding is consistent with our hypothesis that the scarcity of internal resources and lack of investment opportunities during the financial crisis increases the costs of diversification (exacerbation of agency problems) for mutual insurers as they are less monitored than stock insurers. In contrast, we find that the diversification effect for stock insurers is not significantly affected by economic conditions.

We also study the diversification effect for group and non-group insurers under different economic conditions. Similar to our results for mutual insurers, we find that non-group insurers experience a diversification premium under normal economic conditions, but suffer a diversification penalty during the financial crisis. The increased relative value of an internal capital market provided by diversification under normal conditions is higher for non-groups than for group insurers but when internal resources become extremely constrained, this benefit is no longer valuable and as a result, the costs of diversification outweigh the benefits. Similar to our results for stock insurers, the diversification effect for group insurers is not affected by economic conditions.

Our results are robust to alternative measures of performance and diversification, and to corrections for endogeneity. Our findings shed light on the specific economic conditions and ownership structures in which diversification improves or reduces firm performance. The effect of diversification for stock and group insurers is not significantly affected by economic activity. However, economic activity significantly affects the performance of diversified mutual and non-group insurers relative to focused firms with these same ownership structures. Diversification reduces the performance of mutual and non-group insurers during economic downturns. Our study also provides several practical and policy implications. Firms should consider their ownership structure when developing a diversification strategy. Firms should take into account the current and expected state of the economy when deciding whether to diversify or make an adjustment to their current diversification status. This is specifically important for mutual insurers and non-group insurers given that our results show that the performance of these type of ownership structures is significantly affected by economic activity. Finally, firms and regulators should encourage alternative governance mechanisms for mutual and non-group insurers in order to provide more guidance and closer monitoring for these types of ownership structures.

**Author Contributions:** Conceptualization, I.A.L. and Z.L.; methodology, I.A.L. and Z.L.; formal analysis, I.A.L. and Z.L.; investigation, I.A.L. and Z.L.; resources, I.A.L. and Z.L.; data curation, Z.L.; writing—original draft preparation, Z.L.; writing—review and editing, I.A.L. and Z.L.; project administration, I.A.L.

**Funding:** This research received no external funding.

**Conflicts of Interest:** The authors declare no conflict of interest.

## Appendix A. Benefits and Costs of Diversification

**Table A1.** Benefits and Costs of Diversification.

| Benefits | Costs |
|---|---|
| Debt Coinsurance (Lewellen 1971) | Inefficient Investment (Rajan et al. 2000) |
| Efficient Internal Capital Markets (Stein 1997) | Agency Problems: |
| Use of Non-Tradable Resources (Penrose 1959) | Manager's Personal Risk Reduction (Amihud and Lev 1981) |
| Economies of Scope (Teece 1980, 1982) | Empire-building (Jensen 1986) |
| Increased Market Power (Tirole 1995) | Managerial Entrenchment (Shleifer and Vishny 1989) |

## Appendix B. Property/Casualty Insurance Industry Income Analysis, 2004–2016 (1)

PROPERTY/CASUALTY INSURANCE INDUSTRY INCOME ANALYSIS, 2004-2016 (1)

($ billions)

| | 2004 | 2005 | 2006 | 2007 | 2008 | 2009 | 2010 | 2011 | 2012 | 2013 | 2014 | 2015 | 2016 |
|---|---|---|---|---|---|---|---|---|---|---|---|---|---|
| Net written premiums | $424.10 | $425.50 | $443.50 | $440.60 | $434.90 | $418.40 | $422.10 | $438.00 | $456.70 | $477.00 | $496.60 | $514.40 | $528.20 |
| Percent change | 4.90% | 0.30% | 4.20% | -0.60% | -1.30% | -3.80% | 0.90% | 3.40% | 4.30% | 4.40% | 4.10% | 3.50% | 2.70% |
| Earned premiums | $413.80 | $417.60 | $435.50 | $438.90 | $438.30 | $422.30 | $420.50 | $434.40 | $448.90 | $467.40 | $487.60 | $506.00 | $523.50 |
| Losses incurred | 247.8 | 256.5 | 231.3 | 244.7 | 286.3 | 253.8 | 256.5 | 290.8 | 277.7 | 259.4 | 277.4 | 290.7 | 318 |
| Loss adjustment expenses incurred | 53.1 | 55.1 | 52.6 | 52.3 | 51.7 | 52.5 | 52.6 | 53.8 | 55.5 | 55.6 | 57.3 | 59.6 | 60.3 |
| Other underwriting expenses | 106.8 | 109.8 | 117.1 | 120.1 | 119.6 | 117 | 119.6 | 124.2 | 128.9 | 134.6 | 138.1 | 144.3 | 147.6 |
| Policyholder dividends | 1.7 | 1.9 | 3.4 | 2.4 | 2 | 2 | 2.3 | 1.9 | 2.1 | 2.5 | 2.5 | 2.5 | 2.3 |
| Underwriting gain/loss | 4.3 | -5.6 | 31.1 | 19.3 | -21.2 | -3 | -10.4 | -36.2 | -15.4 | 15.2 | 12.3 | 8.9 | -4.7 |
| Investment income | 40 | 49.7 | 52.3 | 55.1 | 51.5 | 47.1 | 47.2 | 49.2 | 48 | 47.3 | 46.2 | 47.2 | 46.3 |
| Miscellaneous income/loss | -0.3 | 1 | 1.2 | -1 | 0.4 | 0.9 | 1 | 2.5 | 2.4 | 1.5 | -2.8 | 1.5 | 1 |
| Operating income/loss | 44 | 45.1 | 84.6 | 73.4 | 30.6 | 45 | 37.8 | 15.4 | 35 | 64.1 | 55.6 | 57.7 | 42.6 |
| Realized capital gains/losses | 9.1 | 9.7 | 3.5 | 8.9 | -19.8 | -7.9 | 5.7 | 7 | 6.2 | 11.4 | 10.1 | 9.4 | 7.3 |
| Incurred federal income taxes/credit | 14.6 | 10.7 | 22.4 | 19.8 | 7.8 | 8.4 | 8.9 | 3 | 6.1 | 12 | 10.2 | 10.2 | 7.3 |
| Net income after taxes | 38.5 | 44.2 | 65.8 | 62.5 | 3 | 28.7 | 34.7 | 19.5 | 35.1 | 63.4 | 55.5 | 56.8 | 42.6 |

(1) Data in this chart exclude state funds and other residual market insurers and may not agree with similar data shown elsewhere from different sources.

Source: ISO®, a Verisk Analytics® business.

**Appendix C. Business Lines Detail**

Consistent with Liebenberg and Sommer (2008), we use Direct Premium Written (DPW) to capture an insurer's underwriting operations and exclude reinsurance line. There are total 34 business lines reported in NAIC on 2004. On 2008, warranty (line 30) is added in NAIC reporting statement. In 2009, excess worker's compensation (line 17.3) is added in NAIC reporting statement. We made the following modifications to combine lines that belong to the same business line. After that, we have 23 business lines in total.

(a)     Fire and Allied lines is defined as the sum of "Fire" (line 1) and "Allied lines" (line 2).

(b)     Accident and Health is defined as the sum of "Group Accident and Health" (line 13), "Credit Accident and Health" (line 14), and "Other Accident and Health" (line 15).

(c)     Medical Malpractice is defined as the sum of "Medical Malpractice–Occurrence" (line 11.1) and "Medical Malpractice–Claims Made" (line 11.2).

(d)     Products Liability is defined as the sum of "Products Liability–Occurrence" (line 18.1) and "Products Liability–Claims Made" (line 18.2).

(e)     Auto is defined as the sum of "Private Passenger Auto Liability" (line 19.1, 19.2), "Commercial Auto Liability" (line 19.3, 19.4), and Auto Physical Damage (line 21).

(f)     Other liability is defined as the sum of "Other liability–occurrence" (line 17.1) and "Other liability–claims-made" (line 17.2).

(g)     Other lines is defined as the sum of "Aggregate write-ins for other lines of business" (line 34), "Excess workers' compensation" (line 17.3) and "Warranty" (line 30).

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
