# Peer review of "The Effect of Diversification under Different Ownership Structures and Economic Conditions: Evidence from the Great Recession"

_jrfm, doi:10.3390/jrfm12020082_

Reviewer 1 Report

referee report attached

The paper studies the mediating effects of insurance company ownership on the relation between diversification and performance under different economic situations. The paper is well written and easy to follow. Below are my comments to further improve it.

Major comments:

In the introduction, you need to strengthen the contributions. You can talk about the followings: What insights can you provide based on your finding? Do they push forward our understanding? What should we do with your research? Do you have any suggestions to improve the current regulation or practice? Adding the above discussion and extend your literature review may help you make more contributions and position your contributions better. One policy implication is that firms and regulators should develop different diversification strategies and encourage alternative governance mechanisms for firms with ownership structures. You can go further in this line and discuss details.

The current paper is weak in addressing/discussing the endogeneity problem. First, firm performance obviously impacts firm diversification, ownership choice, governance mechanisms, etc. For example, when a firm performs well or expects to perform well, it is more likely to expand and diversify. Second, the only method you use to deal with the endogeneity problem is the IV method. Specifically, you use reinsurance rate and index of attractiveness to single-line insurers, which clearly are not (perfectly) exogenous. For example, performance and market attractiveness (even just to single-line insurers) are correlated. They affect the dependent variable firm value, not necessarily through the channel of diversification only. I suggest you use more methods to mitigate the endogeneity problem and show robustness. See Li 2016, Endogeneity in CEO power: A survey and experiment, Investment Analysts Journal, 45 (3): 149-162 for a summary of methods to deal with the endogeneity problem without a good IV.

Related to the above point, you should study and rationalize the use of firm size measures in the literature. See Dang et al. 2018. Measuring Firm Size in Empirical Corporate Finance. Journal of Banking & Finance, 86:159-176.  After all, size is the most significant variable in most studies in this area. You need to justify your measure and talk about alternative measures whenever possible.

Minor comments:

The conclusion section, which is too short, should summarize your findings and implications, reinforce your contributions, and point out caveats and future research directions.

It may be clearer if you list your hypotheses as H1, H2… and refer to them when discussing your results.

You need to proofread the paper and update the reference list.

Author Response

We thank the referee for the careful reading of our paper and their thoughtful comments. We have attempted to address each of the referee’s comments, and we believe that these changes have significantly strengthened the revised paper. Throughout this document, we include the reviewer’s comments (in italics) and provide our specific responses in red.  The changes we made to the original manuscript are shown as part of our responses to each of the referee’s comments and they are also highlighted in yellow in the manuscript. Attached please find the detailed responses.

Reviewer 2 Report

This is a potentially interesting paper, but it needs to be revised significantly.

The pages are not numbered.

The literature review should be expanded.

The development of the hypotheses to be tested should be in the context of a specific model specification.

The sample is from 2004 to 2013 (abstract), or 2004-2009 (section 4.1), which seems a little dated.

The inclusion of the dummy variables is not motivated persuasively.

There are too many tables, and several are difficult to read (such as Tables 2, 4, 5, 6 and 7).

Contrary to what is stated in the last paragraph of the paper, the absence of diagnostic checks makes it difficult to determine the robustness of the empirical results.

Author Response

We thank the referee for the careful reading of our paper and their thoughtful comments. We have attempted to address each of the referee’s comments, and we believe that these changes have significantly strengthened the revised paper. Throughout this document, we include the reviewer’s comments (in italics) and provide our specific responses in red.  The changes we made to the original manuscript are shown as part of our responses to each of the referee’s comments and they are also highlighted in yellow in the manuscript. Attached please find the detailed responses.

Round  2

Reviewer 1 Report

Much improved.

Reviewer 2 Report

The revised version is a significant improvement on the original submission.

As most of the reviewer's comments and recommendations have been satisfied, the paper is suitable for publication.